# Noncovalent antibody catenation on a target surface greatly increases the antigen-binding avidity

Jinyeop Song[1†‡], Bo-Seong Jeong[2†], Seong-Woo Kim[2], Seong-Bin Im[2], Seonghoon Kim[2], Chih-Jen Lai[3], Wonki Cho[2], Jae U Jung[3], Myung-Ju Ahn[4], Byung-Ha Oh[2]*

[1]Department of Physics, Korea Advanced Institute of Science and Technology, Daejeon, Republic of Korea; [2]Department of Biological Sciences, KAIST Institute for the Biocentury, Korea Advanced Institute of Science and Technology, Daejeon, Republic of Korea; [3]Cancer Biology Department, Infection Biology Program, and Global Center for Pathogen and Human Health Research, Lerner Research Institute, Cleveland Clinic, Cleveland, United States; [4]Department of Medicine, Samsung Medical Center, Sungkyunkwan University School of Medicine, Seoul, Republic of Korea

*For correspondence:
bhoh@kaist.ac.kr

[†]These authors contributed equally to this work

Present address: [‡]Department of Physics, Massachusetts Institute of Technology, Cambridge, United States

**Abstract** Immunoglobulin G (IgG) antibodies are widely used for diagnosis and therapy. Given the unique dimeric structure of IgG, we hypothesized that, by genetically fusing a homodimeric protein (catenator) to the C-terminus of IgG, reversible catenation of antibody molecules could be induced on a surface where target antigen molecules are abundant, and that it could be an effective way to greatly enhance the antigen-binding avidity. A thermodynamic simulation showed that quite low homodimerization affinity of a catenator, *e.g.* dissociation constant of 100 μM, can enhance nanomolar antigen-binding avidity to a picomolar level, and that the fold enhancement sharply depends on the density of the antigen. In a proof-of-concept experiment where antigen molecules are immobilized on a biosensor tip, the C-terminal fusion of a pair of weakly homodimerizing proteins to three different antibodies enhanced the antigen-binding avidity by at least 110 or 304 folds from the intrinsic binding avidity. Compared with the mother antibody, Obinutuzumab(Y101L) which targets CD20, the same antibody with fused catenators exhibited significantly enhanced binding to SU-DHL5 cells. Together, the homodimerization-induced antibody catenation would be a new powerful approach to improve antibody applications, including the detection of scarce biomarkers and targeted anticancer therapies.

## Editor's evaluation

This important study documents the use of computational models and protein design to enhance antibody binding. The new method could have a broad and immediate impact on a variety of diagnostic procedures that use antibodies as sensitivity is often an issue in these kinds of experiments. The evidence produced is highly compelling through demonstration of the substantial sensitivity enhancement achieved in two test cases. This manuscript will likely be of interest to researchers who use antibodies for diagnostic and therapeutic purposes.

## Introduction

Immunoglobulin G (IgG) antibodies have become the principal therapeutic biologic. IgG antibodies are a homodimer of a heterodimer composed of two copies of each heavy chain (~50 kDa) and light chain (~25 kDa). They have two functional regions: the antigen-binding fragment (Fab) region at the

N-terminal end and the fragment crystallizable (Fc) region at the C-terminal end. With an overall shape of the letter Y, the two identical regions of Fab form two arms that can bind two antigen molecules. This antibody-antigen engagement could prevent the antigen from binding to cognate partners or eliminate the antigen molecules from the cell surface by receptor-mediated endocytosis (*Liu, 2018*). The two copies of Fc form a homodimeric tail that enables a long half-life via binding to the neonatal Fc receptor (FcRn) and exerts effector functions via binding to the Fcγ receptors on effector immune cells or the complement factor C1q (*Hogarth and Pietersz, 2012*; *Lee et al., 2017*), which could lead to the death of cells to which antibody molecules are bound (*Carter and Lazar, 2018*; *Goydel and Rader, 2021*; *Jiang et al., 2011*).

IgG antibodies have desirable properties for use as a therapeutic drug, including high specificity for a target antigen, low immunogenicity and long serum half-life (*Weiner et al., 2010*). On the other hand, therapeutic monoclonal antibodies (mAbs) show side effects, albeit to a lesser degree in comparison with conventional chemotherapeutics, such as low or high blood pressure and kidney damage (*Hansel et al., 2010*). In the case of targeted cancer therapy, where mAbs target a specific antigen on cancer cells, the side effects likely arise due to the expression of the target antigen not only on cancer cells but also on normal cells, which therefore are targeted indiscriminately by mAbs administered in patients (*Scott et al., 2012*). Moreover, mAbs often suffer from shortcomings such as moderate therapeutic efficacy (resulting in the development of resistance) and their efficacy in a fraction of patients (as observed for mAbs against immune checkpoint inhibitors; *Aldeghaither et al., 2019*; *Hansel et al., 2010*; *Wang et al., 2021*). Insufficient blockade of target antigens for various reasons, including insufficient antigen-binding affinity, could be responsible for the moderate therapeutic efficacy.

In general, diagnostic and therapeutic antibodies are required to exhibit low nanomolar or higher antigen-binding affinity ($K_D$ <10 nM) (*Sliwkowski and Mellman, 2013*). To reach this level of affinity, laborious experiments for affinity maturation are usually followed after an initial discovery of an antibody (*Hoogenboom, 2005*). Increasing the valency of binding interaction could be a method of choice. It was shown that irreversibly dimerized monovalent binders can bind targets significantly better than monomeric counterparts (*Foreman and Vilar, 2017*). This enhancement of the binding affinity arises from the proximity effect, where the binding of one subunit of the dimer to a target restricts the search space for the other subunit. Reversibly dimerized binders could also exhibit significantly enhanced binding affinity depending on the affinity for binder-target interaction, affinity for homodimerization and the length of the connecting linker, as predicted by a reacted-site probability approach (*Foreman and Vilar, 2017*). Such approaches to increase the valency of binding have been applied to IgG antibodies, and a considerable increase in the antigen-binding affinity was observed in vitro (*White et al., 2014*). However, as the size of an IgG-type antibody is large (~150 kDa), irreversible cross-linking or tight reversible dimer formation of the antibody would result in poor solubility and tissue penetration in vivo.

Owing to the overall dimeric structure, IgG antibodies genetically fused to a homodimeric protein at the C-terminus can be catenated in an arm-in-arm fashion as long as the homodimer can be formed, not within an antibody molecule, but between two antibody molecules. In theory, it would be possible to generate a soluble fusion protein that remains monomeric in solution, but becomes catenated by the proximity effect on a cell surface where target antigen molecules are abundant, provided that the fused protein has appropriately low homodimerization affinity. Importantly, this proximity effect-driven catenation, in turn, should result in enhanced bivalent antigen-binding affinity (=avidity). In this work, by agent-based modeling (ABM) and proof-of-concept experiments, we demonstrate that antibody catenation induced by intermolecular homodimerization can enormously enhance the antigen-binding avidity of an antibody on a target surface.

## Results

### The concept of antibody catenation on a target surface

This concept was based on (i) the unique dimeric structure of the IgG-type antibody and (ii) a proximity effect that potentially takes place on a target cell surface. In the structure of IgG, the Fc domain is composed of two copies of the constant regions of the heavy chain ($C_{H2}$ and $C_{H3}$) forming a homodimer, in which the two C-termini are ~23 Å apart and point away from each other (*Figure 1A*, *Left*).

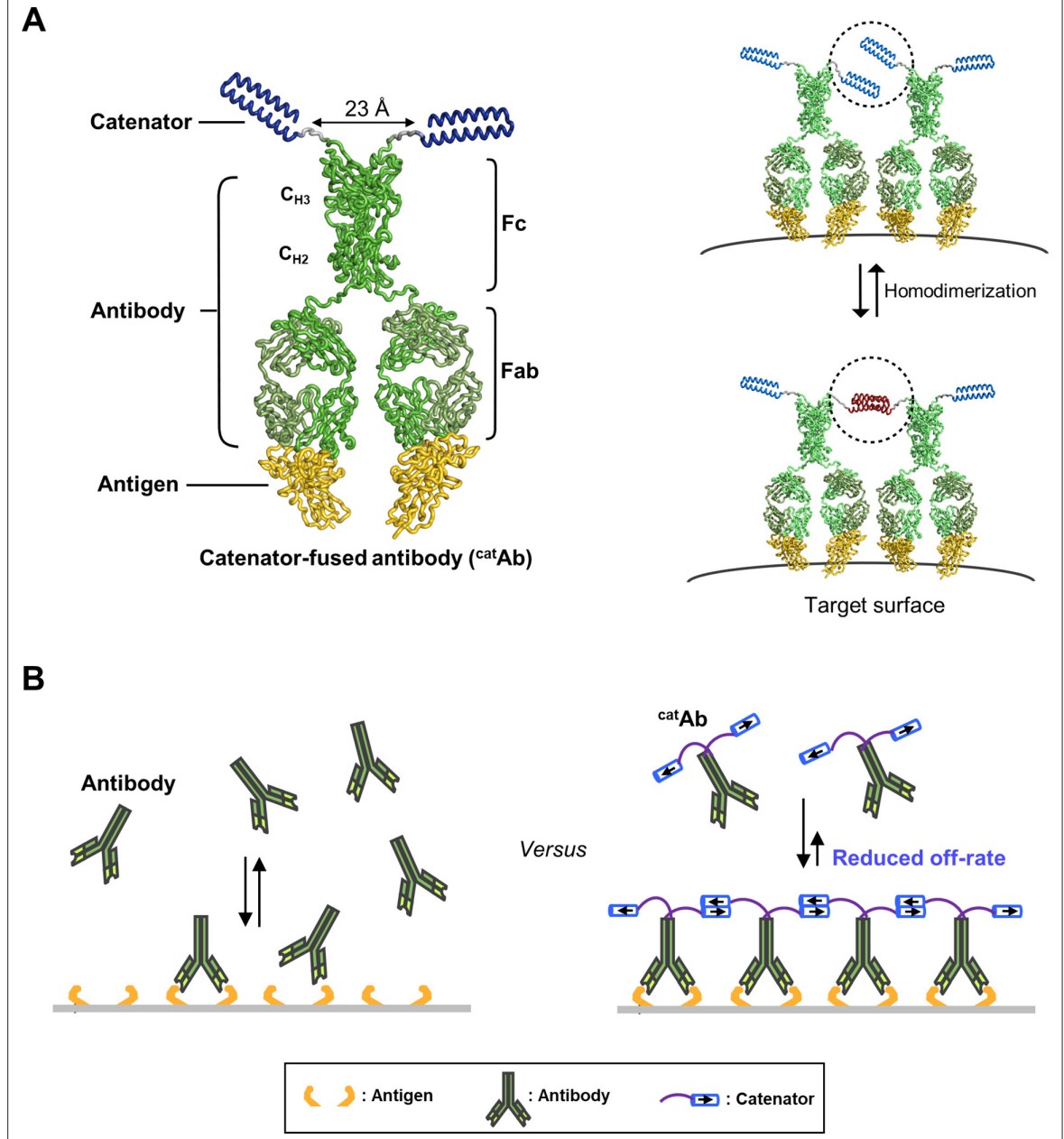

**Figure 1.** The concept of antibody catenation on a target surface by fusion of a catenator. (**A**) Molecular model for catenator-fused antibodies. A flexible linker (Gly-Gly-Ser) between Fc and the catenator and the hinge segment between Fc and Fab were modeled by using the ROSETTA software. The catenator is an α-helical hairpin that forms four-helix anti-parallel coiled coils (PDB entry: 1ROP). The structure of Fc was derived from the IgG1 antibody (PDB entry: 1IGY) and that of Fab from an antibody against the receptor-binding domain of the SARS-CoV-2 spike protein (PDB entry: 6XE1). (**B**) Decreased dissociation by antibody catenation. Pairs of catAb-antigen complexes adjacent to each other can be catenated, and the catAb molecules are increasingly harder to dissociate from each other with increased catenation. The effective antigen-binding avidity would increase owing to a decreased off rate of catAb.

This structural feature indicated that a homodimer-forming protein genetically fused to the C-terminus can be prevented from forming a homodimer intramolecularly by controlling the length of the connecting linker or its homodimerization affinity. Instead, the fusion protein can form a homodimer intermolecularly, and then such a homodimerization could result in a catenation of the antibody molecules (*Figure 1A*, *Right*). We designate the fusion protein between an antibody and a homodimeric protein as antibody-catenator (catAb). A proximity effect for catAb is expected on a target surface where multiple copies of target antigen are present, because the local concentration of catAb on the surface

will increase owing to the antibody-antigen binding interaction. Consequently, the homodimerization between the catenator molecules will increase to form catenated antibodies in an arm-in-arm fashion (*Figure 1B*). Importantly, the effective antigen-binding affinity of $^{cat}$Ab will increase in parallel with the catenation, and the fold enhancement would depend on the degree of the catenation. Thus, it appeared possible to enhance the antigen-binding avidity of the IgG-type antibodies by genetically fusing a weakly homodimer-forming protein.

## Agent-based modeling to simulate the behavior of $^{cat}$Ab

ABM is a computational modeling approach that has been employed in a variety of research areas, including statistical physics (*Perc et al., 2017*; *Fu and Wang, 2008*) and biological sciences (*An et al., 2009*; *Metzcar et al., 2019*; *McLane et al., 2011*). ABM enables the understanding of macroscopic behaviors of a complex system by defining a minimal set of rules governing microscopic behaviors of agents which compose the system.

We constructed an ABM to simulate the behavior of the $^{cat}$Ab molecules on a target surface, where target antigen (Ag) molecules form antibody-binding sites. To circumvent complexity, we presumed that each binding site is a pair of two antigen molecules (2Ag), and $^{cat}$Ab make a bivalent interaction with the binding site in a 1:1 stoichiometry to form an occupied binding site ($^{cat}$Ab-2Ag; *Figure 2A*, *Left*). For catenation to occur between two adjacent $^{cat}$Ab-2Ag complexes, the distance between the centers of two adjacent complexes (*d*) should be closer than the reach length (*L*) defined as *l+c/2*, the sum of the linker length (*l*) and the half the catenator length (*c*) (*Figure 2A*, *Right*). Therefore, multiple parameters affect the catenation on the target surface. In our ABM model, we regarded every possible binding site on the target surface as an individual agent in the ABM formalism, and each binding site is assigned to a fixed position on a three-dimensional (3D) surface with a periodic boundary condition. Three rules in our ABM govern the behaviors of the $^{cat}$Ab molecules on the target surface. The first rule is about the *intrinsic antibody-antigen binding*. An unoccupied binding site binds to one free $^{cat}$Ab through bivalent interaction to form an occupied binding site. Bound $^{cat}$Ab may dissociate from the occupied binding site, leaving the binding site unoccupied. The equilibrium population of the occupied and unoccupied binding sites is determined by the antibody's intrinsic avidity for the antigen with no effect of the catenator on the antigen-binding avidity assumed. Then, the relative likelihood of the occupied state compared to the unoccupied state for any binding site (the likelihood of intrinsic antigen binding) is defined as [$^{cat}$Ab-2Ag]/[2Ag] and thus can be expressed as [$^{cat}$Ab]/$K_D$, where [$^{cat}$Ab] is the concentration of $^{cat}$Ab and $K_D$ is the dissociation constant for the bivalent $^{cat}$Ab-2Ag interaction (*Figure 2B*, *Left*). The second rule is about *catenation*. A pair of $^{cat}$Ab-2Ag complexes on the target surface can be bridged by intermolecular homodimerization between catenators (*Figure 2B*, *Middle*). For a pair of $^{cat}$Ab-2Ag complexes separated by *d* (*Figure 2A*, *Right*), the relative likelihood of the catenation state as compared to the non-catenation state is the ratio of the forward reaction rate (catenation) to the reverse reaction rate (decatenation). The forward reaction rate ($R_{catenation}$) and the reverse reaction rate ($R_{decatenation}$) are given as,

$$R_{catenation} = \left(k_f\right)_{catination} * \left(\frac{1}{N_A} * \frac{1}{V_{sphere}}\right)^2 * V_{overlap}\left(d\right)$$

$$R_{decatenation} = \left(k_r\right)_{catenation} * \frac{1}{N_A}$$

, where $k_f$ and $k_r$ are the reaction rate constant of the forward and reverse reaction, respectively, $N_A$ is the Avogadro number, $V_{sphere}$ is the local spherical volume within the reach of the catenator, and $V_{overlap}(d)$ is the volume where two catenators can come in contact to form a homodimer (*Figure 2A*, *Right*). In approximating the forward reaction rate, the catenator was assumed to sample $V_{sphere}$ uniformly. The relative likelihood, defined as $R_{catenation}/R_{decatenation}$, is then expressed as

$$\frac{R_{catenation}}{R_{decatenation}} = \frac{\left(k_f\right)_{catenator}}{\left(k_r\right)_{catenator}} * \frac{1}{N_A} * \left(\frac{1}{V_{sphere}}\right)^2 * V_{overlap}\left(d\right) = \frac{f(d)}{\left(K_D\right)_{catenator}}$$

where

$$f\left(d\right) = \frac{1}{N_A} * \left(\frac{1}{V_{sphere}}\right)^2 * V_{overlap}\left(d\right)$$

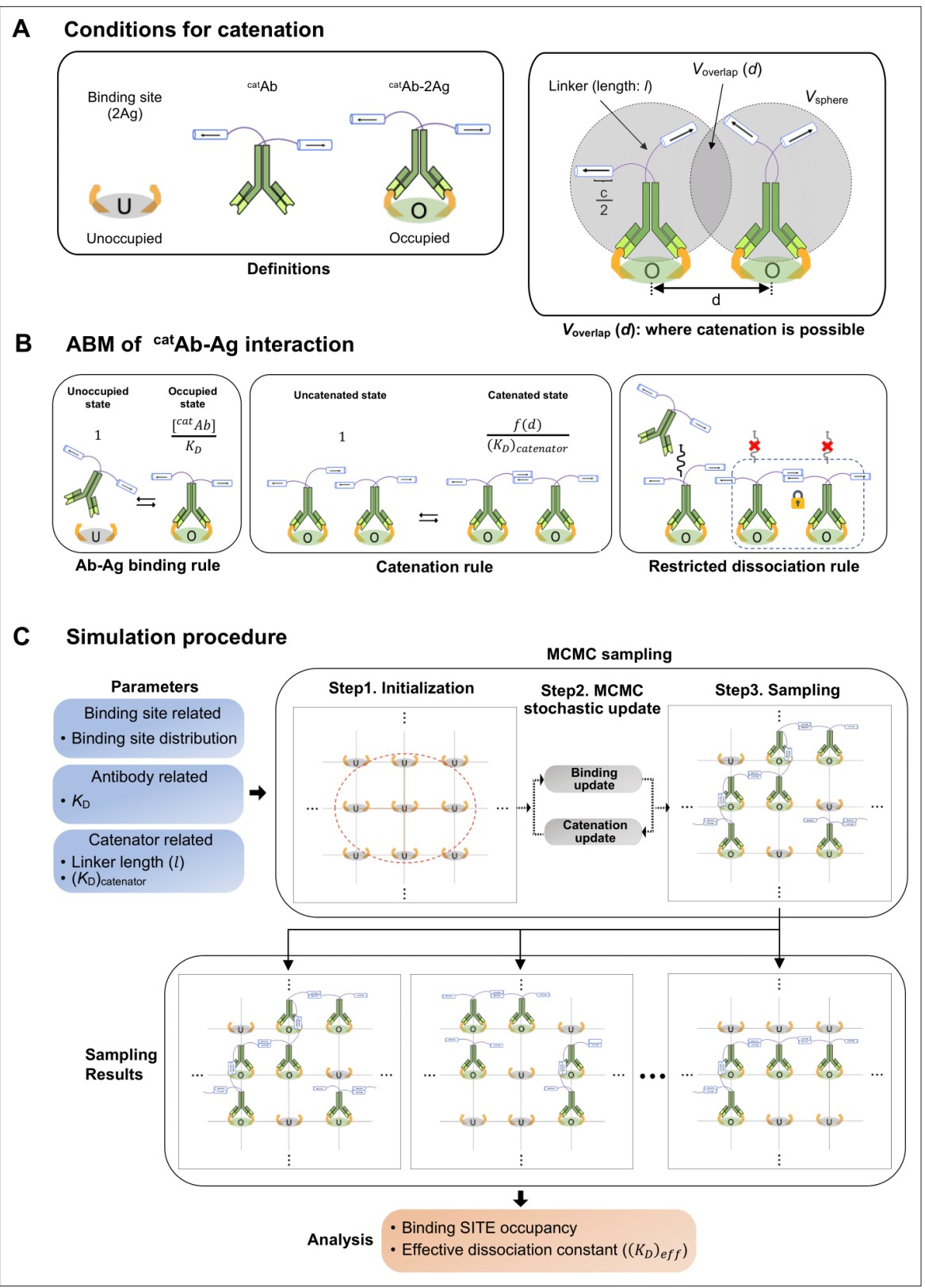

**Figure 2.** ABM for simulating the binding dynamics of a catenator-fused antibody. (**A**) (*Left*) Each binding site is composed of two antigen molecules (2Ag). (*Right*) The gray circles indicate the sphere sampled by the catenator, and $V_{overlap}$ is the overlapping volume between the adjacent spheres. Catenation between two $^{cat}$Ab molecules is possible only in $V_{overlap}$. (**B**) The three rules of the ABM model. (*Left*) $^{cat}$Ab-2Ag binding occurs with a relative likelihood, $[^{cat}$Ab]/$K_D$. (*Middle*) The catenation between adjacent $^{cat}$Ab-2Ag complexes occurs with an indicated relative likelihood, $f$(d)/$(K_D)_{Catenator}$.

*Figure 2 continued on next page*

*Figure 2 continued*

determined by $(K_D)_{Catenator}$ and the inter-complex distance $d$. (*Right*) It was assumed that $^{cat}$Ab molecules that are catenated do not dissociate from the surface. (**C**) The simulation requires specification of the parameters for the binding site, antibody and catenator. Through the MCMC sampling, the state of binding sites on the target surface is iteratively updated with the ABM rules and eventually sampled. A sufficient number of sampling results are collected to quantify the binding occupancy and the effective dissociation constant.

The online version of this article includes the following source data and figure supplement(s) for figure 2:

**Figure supplement 1.** Calculation of $f(d)$ using uniform local density approximation.

**Figure supplement 1—source data 1.** Simulation data for catenation enhancement by binding sites interval and linker length in *Figure 2—figure supplement 1*.

The relative likelihood is thus a function of $d$, and it is inversely proportional to the dissociation constant of the catenator in the bulk medium, $(K_D)_{catenator}$. The function $f(d)$ can be viewed as the effective local concentration of the catenator in $V_{overlap}(d)$. As expected, $f(d)$ and thus the relative likelihood is sensitively affected by the reach length and limited by the $^{cat}$Ab-$^{cat}$Ab distance (*Figure 2—figure supplement 1*). Finally, the third rule is about *restricted dissociation* which assumes that catenated antibodies are not allowed to dissociate from the binding site, because the catenated arms would hold the dissociated antibody near its binding site, forcing it to rebind immediately (*Liese and Netz, 2018*). Under this assumption, antibody molecules are allowed to dissociate from the binding site, only if its catenator is not engaged in the homodimerization with nearby $^{cat}$Ab-2Ag complexes (*Figure 2B*, *Right*).

## Simulations show significant enhancement of the antigen-binding avidity

According to the postulated rules, we simulated the effects of the antibody catenation on the binding interaction between $^{cat}$Ab and 2Ag on a three-dimensional surface by using the Markov Chain Monte-Carlo (MCMC) sampling method (*Hooten and Wikle, 2010*) (see Methods section). Our sampling procedure is composed of three steps (*Figure 2C*). The first step is an *initialization*, where a target surface with the antibody-binding sites is defined by specifying the coordinates for each site. A set of binding sites are positioned equidistant from each other or randomly positioned, and the inter-site distance or the number of binding sites were set as variables. The next step is an *MCMC stochastic update* step. In each iteration, a binding site is randomly selected from the target surface, and the probability of changing the status of the selected binding site (occupied or not) is calculated by the Metropolis-Hasting algorithm (*Hastings, 1970*; *Grazzini et al., 2017*). Then, the 'on' or 'off' state of this site is updated with the calculated probability. Accordingly, the catenation state is probabilistically updated for each update step. In the following sampling step, the total number of the occupied binding sites is counted, which is then collected through multiple simulation runs for the statistical analysis of the binding site occupancy and the effective antigen-binding avidity. The binding site occupancy is the mean value of the number of occupied binding sites collected for more than 1024 MCMC samplings. For each simulation, we calculated the mean binding occupancy and the effective dissociation constant, $(K_D)_{eff}$, which takes into account the effect of the antibody catenation, and is expressed as:

$$\left(K_D\right)_{eff} = \frac{\left(1 - Binding\ Site\ Occupancy\right) * \left[^{cat}Ab\right]}{Binding\ Site\ Occupancy}$$

Since the catenator homodimerization should be affected by how the binding sites are distributed on a 3D surface, simulations were conducted for different arrays of binding sites. In the simulations, $(K_D)_{catenator}$ was the main variable, while other parameters were set constant. First, we simulated the binding sites forming a square lattice to find that the Ab-catenator exhibited enhanced binding site occupancy in a sigmodal manner, and that it could be enhanced to near full saturation by a catenator that forms a homodimer with quite low binding affinity. For instance, a $^{cat}$Ab with $(K_D)_{catenator}$ of ~1 µM exhibited ~70-fold enhancement of the effective antigen-binding avidity (=reduction of $(K_D)_{eff}$) in comparison with the same antibody without a fused catenator (*Figure 3*). As a means of comparison across different simulation setups, we employed '$(K_D)_{catenator,50}$' which is defined as the $(K_D)_{catenator}$ that enables half-maximal enhancement of the binding site occupancy (*Figure 3*).

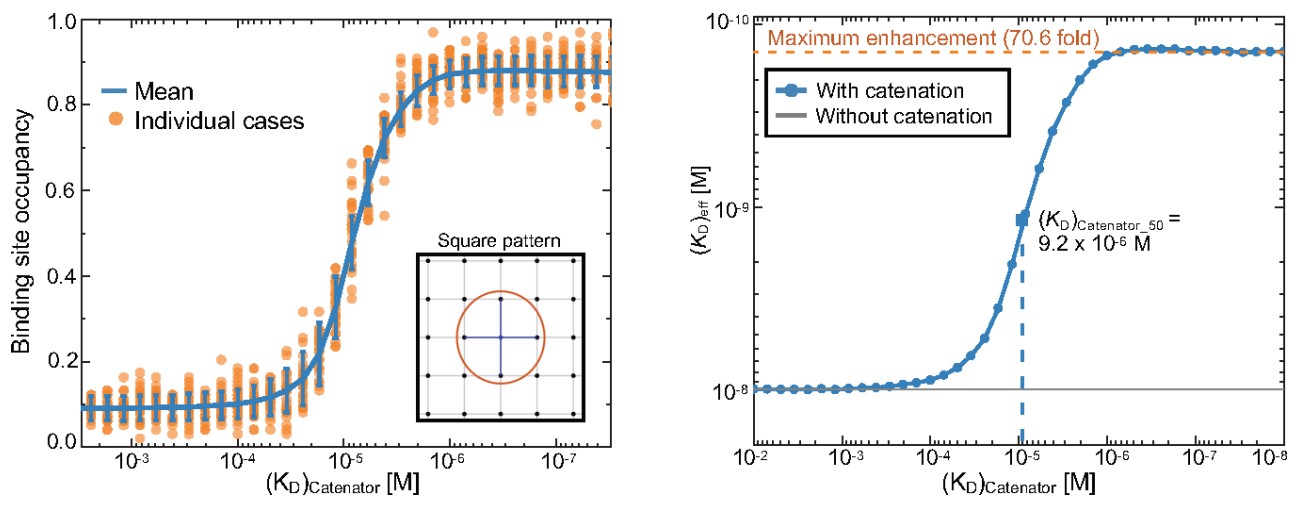

**Figure 3.** Simulations of the binding site occupancy and $(K_D)_{eff}$ in response to $(K_D)_{catenator}$. (*Left*) Binding site occupancy. The simulations were carried out for a square array of the binding sites. The values for a set of variables were $K_D = 10^{-8}$ M, $[^{cat}Ab]=10^{-9}$ M, reach length = 7 nm, spacing between the binding sites = 12 nm and the number of total binding sites = 98. The mean value and standard deviations of 1024 MCMC simulations for each $(K_D)_{catenator}$ value are shown in blue, and the data are shown as a scatter plot of representative runs (orange). (*Right*) The effective dissociation constant. The data shown on the left were converted into the $(K_D)_{eff}$ values. The dashed line represents the $K_D$ value for the same antibody without a catenator. The maximum fold enhancement of the effective binding avidity, which is equivalent to the reduction of $(K_D)_{eff}$, is 70.6.

The online version of this article includes the following source data for figure 3:

**Source data 1.** Simulation data for *Figure 3*.

## Comparison of the simulations for different arrays of the binding sites

Next, we carried out simulations for other regular arrays of the binding sites and for randomly distributed binding sites. Depending on the pattern of regularly distributed binding sites, the number of possible catenations for a given binding site (designated as connectivity number) varies: 3, 4 and 6 for a hexagonal, square or triangular array of the binding sites, respectively (*Figure 4A*). These three arrays showed varying but similar enhancement of the binding site occupancy and the effective antigen-binding avidity by the catenator (*Figure 4A*). As expected, the higher the connectivity number was, the lower $(K_D)_{catenator,50}$ an array exhibited; the $(K_D)_{catenator,50}$ was 8.0, 9.2 and 12.2 μM for the hexagonal, square and triangular array of the binding sites, respectively. The simulations showed that, as the connectivity number increased, the effective antigen-binding avidity increased with the maximum 41-, 73-, and 93-fold enhancement for the triangular, square and hexagonal array, respectively (*Figure 4A*). Thus, regardless of the distribution patterns, the effective antigen-binding avidity could be increased by at least 41 folds under the simulation conditions where the target surface contains only 98 binding sites.

For the case of randomly distributed binding sites on a 3D surface, which is relevant to target antigen distribution on cell surfaces, we introduced the binding site density ($\rho$), the number of binding sites per unit area which is set to the square of the reach length (7 nm; *Figure 4B*). In the simulations, the total surface area was 5760 nm², and the number of binding sites was 15, 30, 45, 90, or 120, which correspond to the $\rho$ of 1.47, 2.94, 4.41, 8.82, or 11.76. Denser binding sites would increase the connectivity number for a given binding site. As expected, simulations showed that higher binding site density resulted in a higher level of binding site saturation and a much more significant increase in the effective antigen-binding avidity; the maximum fold enhancement ranged from 15 ($\rho$ =1.47)–1062 ($\rho$ =11.76). Likewise, significantly different $(K_D)_{catenator,50}$ values were observed: for example, $4.2\times10^{-6}$ M at the $\rho$ of 11.76 vs $74\times10^{-6}$ M at the $\rho$ of 1.47 (*Figure 4B*). The maximal saturation and the onset $(K_D)_{catenator}$, which begins to exert the catenation effect, were also considerably different. Thus, the catenation effects are sensitively affected by the binding site density, in contrast with the all-or-none catenation effect observed for the regular arrays of the binding sites (*Figure 4B*). In particular, the catenation-induced enhancement of the antigen-binding avidity was remarkably and sensitively affected by the $(K_D)_{catenator}$ values at high binding site density ($\rho$ >4.41;

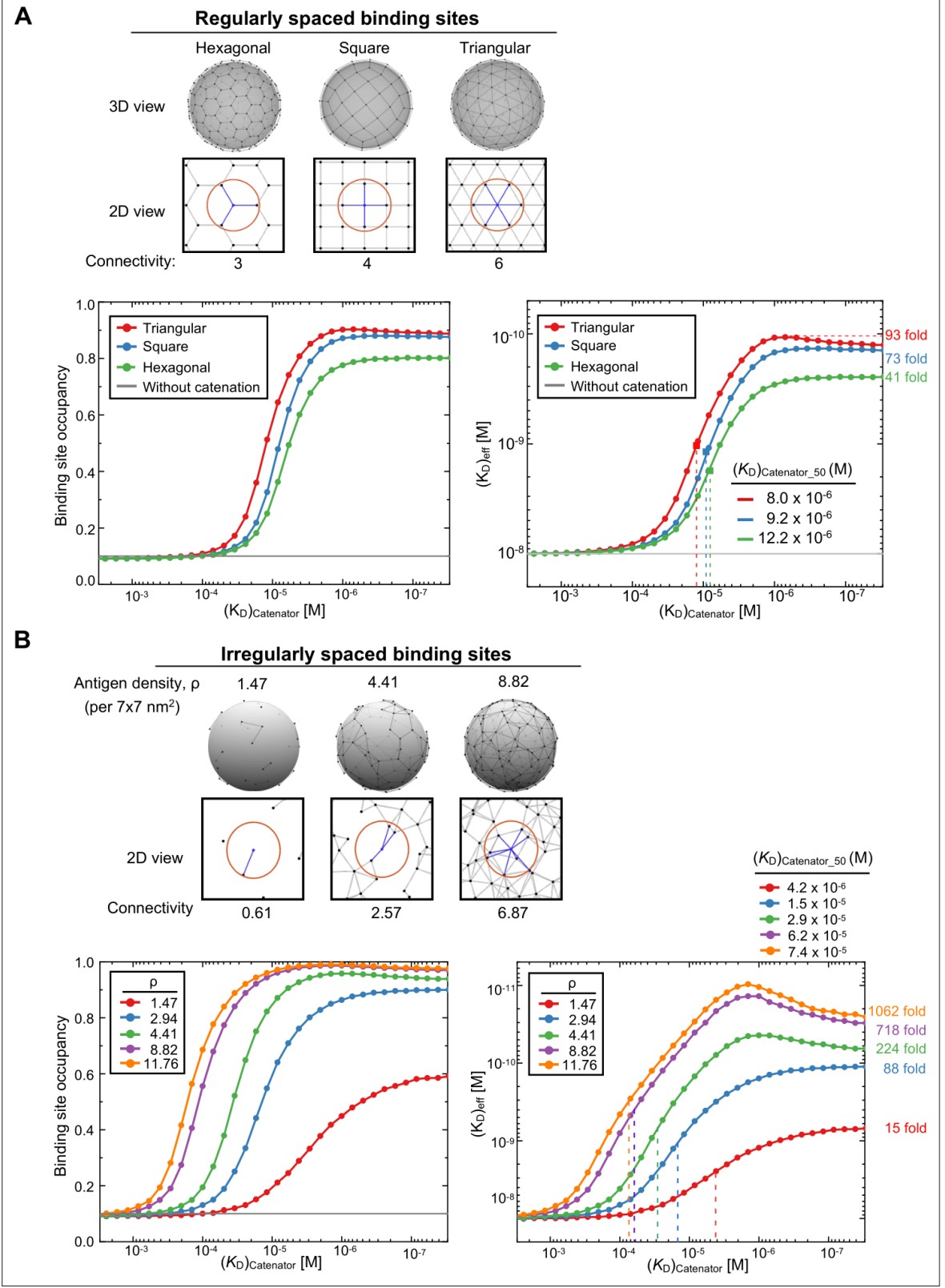

**Figure 4.** Simulations for different arrays of the binding sites. (**A**) Comparison for regularly distributed binding sites. Three different regular arrays of the binding sites are shown at the top. The black dots represent the binding sites and the gray lines the connectable pairs by the catenators. The red circles and the blue lines represent the maximum range of catenation and the connectivity number, respectively, for a given binding site. Binding site occupancy and $(K_D)_{eff}$ in response to $(K_D)_{catenator}$ are shown at the bottom. 1024 trials were sampled for each $(K_D)_{catenator}$ value and the results are plotted.

*Figure 4 continued on next page*

*Figure 4 continued*

The variables were $K_D = 10^{-8}$ M, [$^{cat}$Ab]=$10^{-9}$ M, reach length = 7 nm, spacing between the binding sites = 12 nm, and the number of total binding sites were 98 for the square array and 102 for hexagonal and triangular array, respectively. The numbers on the right are the maximum fold enhancement of the effective binding avidity for each array. (**B**) Comparison for randomly distributed binding sites. Three random arrays of the binding sites with different binding site densities ($\rho$) are shown at the top. The surface area for the simulation was 5760 nm². The simulation conditions were the same as in (**A**). Binding site occupancy and $(K_D)_{eff}$ in response to $(K_D)_{catenator}$ are plotted as in (**A**).

The online version of this article includes the following source data and figure supplement(s) for figure 4:

**Source data 1.** Simulation data for regularly spaced binding sites in *Figure 4*.

**Source data 2.** Simulation data for irregularly spaced binding sites in *Figure 4*.

**Figure supplement 1.** Simulations for randomly distributed, high-density binding sites.

**Figure supplement 1—source data 1.** Simulation data for binding sites with high density in *Figure 4—figure supplement 1*.

**Figure supplement 1—source data 2.** Values of the simulation data for binding sites with high density in *Figure 4—figure supplement 1*.

**Figure supplement 2.** Influence of the likelihood of intrinsic antigen binding ([$^{cat}$Ab]/$K_D$) on binding site occupancy and $(K_D)_{eff}$.

**Figure supplement 2—source data 1.** Simulation data for varying likelihood of intrinsic antigen binding in *Figure 4—figure supplement 2*.

**Figure supplement 3.** Influence of [$^{cat}$Ab]/$K_D$ on binding site occupancy and $(K_D)_{eff}$.

*Figure 4B*). Much greater enhancement was observed as we further increased the density of randomly distributed binding sites:~29,000 maximum fold enhancement at the $\rho$ of 58.8 (*Figure 4—figure supplement 1*), which roughly corresponds to two-hundredths of the density of the HER2 receptor on HER2-overexpressing breast cancer cells (*Peckys et al., 2019*). Together, our simulations show that randomly distributed binding sites at high density enormously enhance the effective antigen-binding avidity of $^{cat}$Ab.

Additionally, we performed simulations for different values of [$^{cat}$Ab]/$K_D$ to estimate the effect of $K_D$ with respect to [$^{cat}$Ab]. Varying [$^{cat}$Ab]/$K_D$ from 0.01 to 1.0 (by varying $K_D$ from $10^{-8}$ to $10^{-6}$) resulted in 85- to 900-fold enhancement of the antigen-binding avidity, suggesting that the catenation effect works for a broad range of $K_D$ values (*Figure 4—figure supplement 2*). We also performed simulations to estimate the effect of [$^{cat}$Ab]. Varying [$^{cat}$Ab] from 0.1 to 10 nM resulted in far less than a 100-fold increase of $(K_D)_{eff}$, showing that the enhancement of $(K_D)_{eff}$ by increasing the concentration of $^{cat}$Ab is much less dramatic than that by increasing the affinity for catenator homodimerization (*Figure 4—figure supplement 3*).

## Proof-of-concept experiments

For experimental validation, we chose stromal cell-derived factor 1α (SDF-1α) as a catenator. SDF-1α is a small (Mr = 8 kDa) and weakly homodimerizing protein ($K_D$ = 150 μM; *Veldkamp et al., 2005*). Initially, by using a 10-residue connecting linker, (G₄S)₂, SDF-1α was fused to the homodimeric heavy chain (H chain) of two different antibodies: Trastuzumab(N30A/H91A), a variant of the clinically used anti-HER2 antibody Trastuzumab and a germline-encoded glCV30, an antibody against the receptor-binding domain (RBD) of the severe acute respiratory syndrome coronavirus 2 (SARS-CoV-2) spike protein. The Fab fragment of Trastuzumab(N30A/H91A) binds to the ectodomain of HER2 with a $K_D$ of 353 nM (*Slaga et al., 2018*), and glCV30 binds to the RBD with a similar binding affinity (not avidity, $K_D$ = 407 nM; *Hurlburt et al., 2020*). Both antibody-catenator constructs exhibited drastic enhancements of the $(K_D)_{eff}$ for the target proteins immobilized on the biosensor surface. However, as described below in the 'incidental observations' section, we now believe that these constructs likely formed reversible heterogeneous oligomers in solution, and the resulting data would not be a correct reflection of the effect of antibody catenation.

It was possible to obviate the oligomerization problem by dually fusing a pair of two different catenators, SDF-1α and SAM, to the knobs-into-holes heterodimeric Fc (HetFc) (*Leaver-Fay et al., 2016*), via a (G₄S)₂ linker. The sterile alpha motif (SAM) domain of SLy1 is also a small (Mr = 7.4 kDa) and weakly homodimerizing protein ($K_D$=117 μM; *Kukuk et al., 2019*). Fusion of the homodimeric proteins to HetFc would also result in the antibody catenation on a target surface in a fashion shown in *Figure 5A*, while their catenation efficiency would be reduced compared with a corresponding homodimeric antibody-catenator construct. Employing a HetFc is instrumental to generate a control antibody, which has a single catenator. This construct can at best form a dimer, but cannot be catenated (*Figure 5A*; Right).

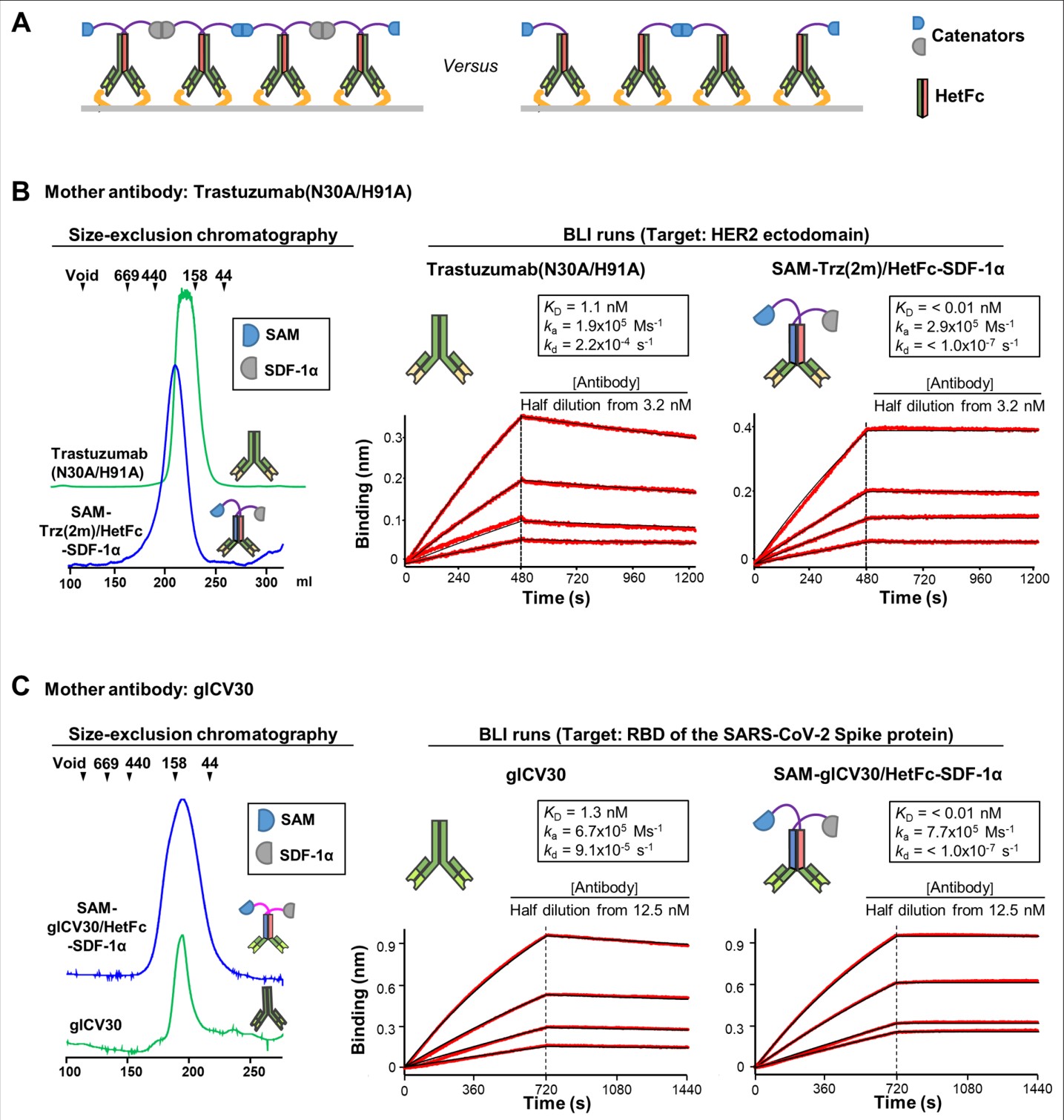

**Figure 5.** Catenation effect on the binding avidity. (**A**) By employing a heterodimeric Fc (HetFc), an antibody fused to two different catenators (*Left*) or an antibody with one catenator arm (*Right*) can be generated. Only the former could form catenated antibodies on a surface where target antigens are abundant. (**B**) Elution profile and BLI analyses of Trastuzumab(N30A/H91A) and SAM-Trz(2 m)/HetFc-SDF-1α. Trz(2 m)/Het stands for Trastuzumab(N30A/H91A) with its homodimeric Fc replaced by a HetFc. The two heavy chains were fused to SAM or SDF-1α. The two antibodies were eluted from a size-exclusion column as if they were monomeric (*Left*). Elution positions of the size marker proteins are indicated by triangles. BLI runs for the two antibodies are shown (*Right*). The ectodomain of HER2 was immobilized on the biosensor tip. (**C**) Elution profile and BLI analyses of glCV30 and SAM-glCV30/HetFc-SDF-1α. The two antibodies were analyzed as described in (**B**). Commonly for all BLI runs, the experimental signals and fitted curves

*Figure 5 continued on next page*

*Figure 5 continued*

are shown in red and black, respectively. For curve fitting, 1:1 binding was assumed. The kinetic parameters are shown in the insets. $k_a$, association rate constant; $k_d$, dissociation rate constant. The experiments were performed in triplicates, and representative sensorgrams are shown.

By replacing the original heavy chain with two heavy chains containing the 'knobs' or 'holes' mutations fused to SAM or SDF-1α, we first generated SAM-Trastuzumab(N30A/H91A)/HetFc-SDF-1α (in short, SAM-Trz(2 m)/HetFc-SDF-1α) and SAM-glCV30/HetFc-SDF-1α (*Figure 5B and C*). While the heterodimer formation was not 100%, it was possible to purify the heterodimeric antibodies homogenously that exhibited a single elution peak from a size-exclusion column that corresponds to the size of the heterodimer (*Figure 5B and C*). Trastuzumab(N30A/H91A) and SAM-Trz(2 m)/HetFc-SDF-1α exhibited similar association kinetics in bio-layer interferometry (BLI) experiments: the association rate constants ($k_a$s) of the two antibodies were $1.9 \times 10^5$ Ms$^{-1}$ and $2.9 \times 10^5$ Ms$^{-1}$, respectively. In contrast, the dissociation rate constants ($k_d$s) of the two antibodies were significantly different: $2.2 \times 10^{-4}$ Ms$^{-1}$ *versus* $<1.0 \times 10^{-7}$ Ms$^{-1}$. As a result, SAM-Trz(2 m)/HetFc-SDF-1α exhibited the $K_D$ less than 0.01 nM (the limit of the instrumental sensitivity), at least 110-fold higher binding avidity compared with that of Trastuzumab(N30A/H91A) ($K_D$ of 1.1 nM; *Figure 5B*). These observed kinetics are consistent with the expectation that fused catenators would not affect the association of the antibodies, but would slow down the dissociation of the antibodies into the bulk solution via catenation of the antibody molecules on the sensor tip. The glCV30 and SAM-glCV30/HetFc-SDF-1α pair also exhibited similar association and dissociation kinetics, and the catenator fusion increased the binding avidity by at least 130 folds (*Figure 5C*).

To prevent the avidity enhancement beyond the instrumental sensitivity limit, we constructed triply mutated Trastuzumab(N30A/H91A/Y100A)/HetFc, which exhibited reduced binding avidity ($K_D$ of 243 nM for HER2) (*Figure 6A*; *Bottom Left*). The two catenator arms were fused to this antibody, and mScarlet to the light chain (for later cell-based experiments) to generate SAM-Trastuzumab(N30A/H91A/Y100A)/HetFc-SDF-1α (L-mScarlet) (in short, SAM-Trz(3 m)/HetFc-SDF-1α (L-mScarlet)). For a control experiment, SAM-Trastuzumab(N30A/H91A/Y100A)/HetFc (L-mScarlet) was generated that contains a single catenator arm. The two antibody-catenator constructs behaved as if they were monomeric in solution (*Figure 6A*, *Top*). BLI analyses showed that Trastuzumab(N30A/H91A/Y100A) and SAM-Trz(3 m)/HetFc-SDF-1α (L-mScarlet) exhibited similar association kinetics, but significantly different dissociation kinetics. As a result, SAM-Trz(3 m)/HetFc-SDF-1α (L-mScarlet) exhibited the $K_D$ of 0.8 nM, 304-fold higher binding avidity compared with that of Trastuzumab(N30A/H91A/Y100A) (*Figure 6A*, *Bottom Middle*). In support of the view that the observed avidity enhancement arises from the catenation of antibody molecules, SAM-Trz(3 m)/HetFc (L-mScarlet), which has one catenator arm and thus cannot catenate the antibody molecules (*Figure 5A*), exhibited high dissociation kinetics and the $K_D$ of 373 nM, which are close to those of the mother antibody (*Figure 6A*, *Bottom Right*).

Another antibody with intentionally reduced binding avidity was constructed: Obinutuzumab(Y101L), a mutant version of the clinically used anti-CD20 antibody (*Evans and Clemmons, 2015*). In our measurement, Obinutuzumab(Y101L) exhibited the $K_D$ of 30.4 nM for CD20 (*Figure 6B*). Also generated was SAM-Obinutuzumab(Y101L)/HetFc-SDF-1α (in short, SAM-Obz(Y101L)/HetFc-SDF-1α), which was monomeric in solution (*Figure 6—figure supplement 1*). This antibody/antibody-catenator pair also exhibited similar association kinetics, but significantly different dissociation kinetics: $k_a$s were $3.3 \times 10^5$ Ms$^{-1}$ *vs* $1.8 \times 10^5$ Ms$^{-1}$, and $k_d$s were $1.0 \times 10^{-2}$ Ms$^{-1}$ *vs* $2.3 \times 10^{-5}$ Ms$^{-1}$ (*Figure 6B*, *Top*). As a result, SAM-Obz(Y101L)/HetFc-SDF-1α increased the binding avidity by 234 folds in comparison with the mother antibody. We asked whether the enhanced binding avidity observed for SAM-Obz(Y101L)/HetFc-SDF-1α could be reproduced in binding to SU-DHL5, a cell line derived from diffuse large B-cell lymphoma that overexpresses CD20. Flow cytometric analysis showed that SAM-Obz(Y101L)/HetFc-SDF-1α bound to the SU-DHL5 cell much more effectively than Obinutuzumab(Y101L) (*Figure 6B*, *Bottom*). This result indicates that the SAM-Obz(Y101L)/HetFc-SDF-1α molecules were likely catenated on the surface of the SU-DHL5 cell. Together, these data demonstrate that noncovalent antibody catenation on a target surface is a way to greatly enhance antigen-binding avidity.

## Incidental observations

The homodimeric glCV30-SDF-1α construct showed three eluted fractions from a size-exclusion column: a void, a broad peak, and a narrow peak fraction (*Figure 7A*). The narrow peak fraction was

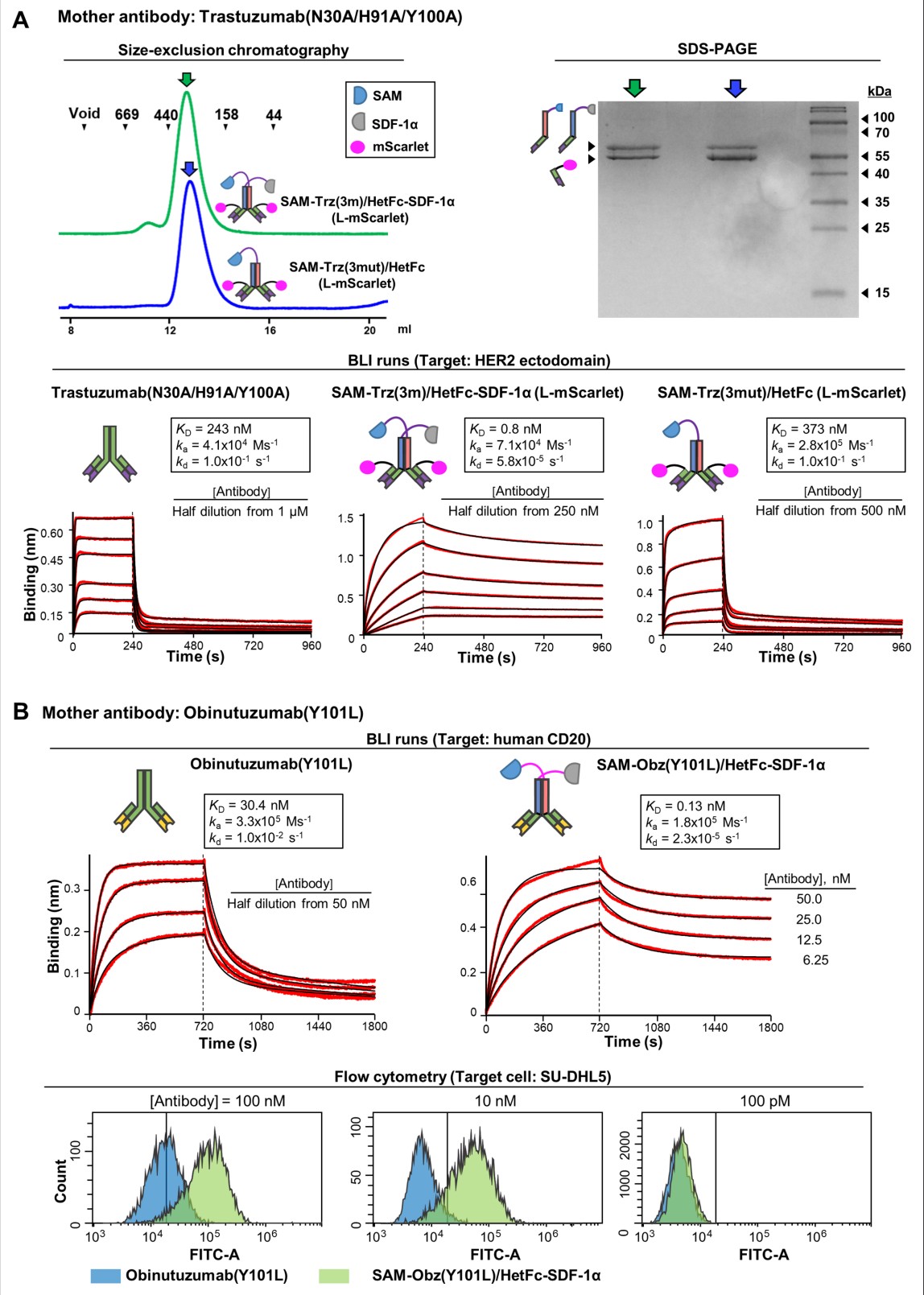

**Figure 6.** Catenation effects were observed for low-affinity mother antibodies. (**A**) The triply mutated Trastuzumab(N30A/H91A/Y100A), the antibody with two catenator arms (SAM-Trz(3 m)/HetFc-SDF-1α) and the same antibody with one catenator arm (SAM-Trz(3 m)/HetFc) were prepared with their light chain (L) fused to mScarlet at the C-terminus. The two antibody-catenators were eluted from a size-exclusion column as if they were monomeric (*Left*). The SDS-PAGE of the peak fractions shows the correct sizes of the indicated chains (*Right*). BLI was performed similarly as described in *Figure 5B*.

*Figure 6 continued on next page*

*Figure 6 continued*

The two catenator-armed antibody exhibits a significant increase in the binding avidity, but the one catenator-armed antibody did not ($K_D$ values in the inset). (**B**) Obnutuzumab(Y101L) and SAM-Obz(Y101L)/HetFc-SDF-1α were prepared, and BLI was performed with biotinylated human CD20 immobilized on the biosensor tip (*Top*). Flow cytometric analyses of SU-DHL5-binding by the two antibodies were performed (*Bottom*). For this experiment, the two antibodies were labeled with muGFP at the C-terminus of the light chain. The fluorescent signal from muGFP was detected by using a 525/40 bandpass filter.

The online version of this article includes the following source data and figure supplement(s) for figure 6:

**Source data 1.** SDS-PAGE result for SAM-Trastuzumab(N30A/H91A/Y100A)/HetFc-SDF-1a and SAM-Trastuzumab(N30A/H91A/Y100A)/HetFc.

**Figure supplement 1.** Purification of SAM-Obz(Y101L)/HetFc-SDF-1ᵅ.

**Figure supplement 1—source data 1.** SDS-PAGE result for SAM-Obinutuzumab(Y101L)/HetFc-SDF1a.

a mixture of the L chain and H chain missing the catenator as a result of partial proteolytic cleavage. In contrast, the broad peak did not contain the cleaved H chain. The broad peak fraction was highly soluble, as it could be concentrated at least to 200 μM. Assuming that the broad peak fraction was in a dynamic equilibrium between monomeric and oligomeric species in solution as a result of the weak intermolecular interaction of SDF-1α, we used it to assess their ability to enhance the binding avidity for the target antigens. glCV30-SDF-1α also exhibited association-dissociation kinetics similar to those of the dual catenator-fused antibodies: a similar $k_a$ but very low $k_d$ in comparison with those of the mother antibody (*Figure 7*). glCV30-SDF-1α exhibited virtually no dissociation and enhanced the binding avidity by at least 5120 folds (*Figure 7B*).

Consistently, the glCV30-SDF-1α construct exhibited notable activity in a virus neutralization assay using vesicular stomatitis virus (VSV) pseudotyped with the SARS-CoV-2 Spike protein. This virus is expected to harbor multiple copies of the Spike protein on its envelope. In comparison with glCV30, glCV30-SDF-1α exhibited a~15 fold lower inhibition constant 50 ($IC_{50}$) value. This neutralization potency of glCV30-SDF-1α ($IC_{50}$ of 0.21 μg/ml) is comparable to that ($IC_{50}$ of 0.25 μg/ml) of CV30, which is an affinity-matured version of glCV30 (*Figure 5C*). CV30 bound to the RBD domain with the $K_D$ of less than 10 pM in our measurement by BLI, similarly as glCV30-SDF-1α did (*Figure 7C*).

We believe that the avidity enhancement and the strong neutralization activity of glCV30-SDF-1α reflect not only the catenation effect but also reversible oligomerization in solution, which further increases the binding avidity. Likely, some fraction of SDF-1α underwent a structural change at the low pH step in the protein purification, leading to reversible heterogeneous oligomerization. Although it is yet unclear why the reversible oligomerization, if this is the case, didn't enhance the association rate constant, these incidental observations suggest a possible approach for producing noncovalent IgG antibody multimers that could enhance the sensitivity of diagnostic antibodies.

## Discussion

In the current phage display for antibody screening, many candidates that do not satisfy a required affinity for a target antigen are rejected, although they might have high specificity of binding. A simple and general way of increasing the antigen-binding affinity of antibodies would be highly valuable for various applications of antibodies. Taking advantage of the particular homodimeric structure of IgG antibodies, we put forth a concept to enhance the bivalent antigen-binding interaction by fusing a weakly homodimerizing protein to the C-terminus of Fc. The validity of the concept was tested by simulations based on an ABM and supported by experimental demonstrations.

Our ABM with the three postulated rules was the basis for predicting the enhancement of effective antigen-binding avidity. The model has caveats. First, the assumption of uniform density for the fused catenators within a sphere oversimplifies the dynamics of the catenators, which would highly depend on physical contexts, such as molecular orientations and potential intramolecular interaction with the antibody (*Zhou, 2001*). Second, the binding sites representing antigens are fixed on a surface in our model, but in real situations, antigens move their positions, *e.g.*, receptor molecules on cellular membranes (*Saxton and Jacobson, 1997*). Advanced molecular dynamics simulations incorporated into the ABM would take account of these microscopic details to result in a more accurate prediction of the behaviors of the ᶜᵃᵗAb molecules and the binding sites. Despite these caveats, the simulations provided valuable insights into the proper ranges of the antigen-binding avidity of an antibody and catenator-catenator binding affinity. According to the simulation, we adopted SDF-1α and SAM, as

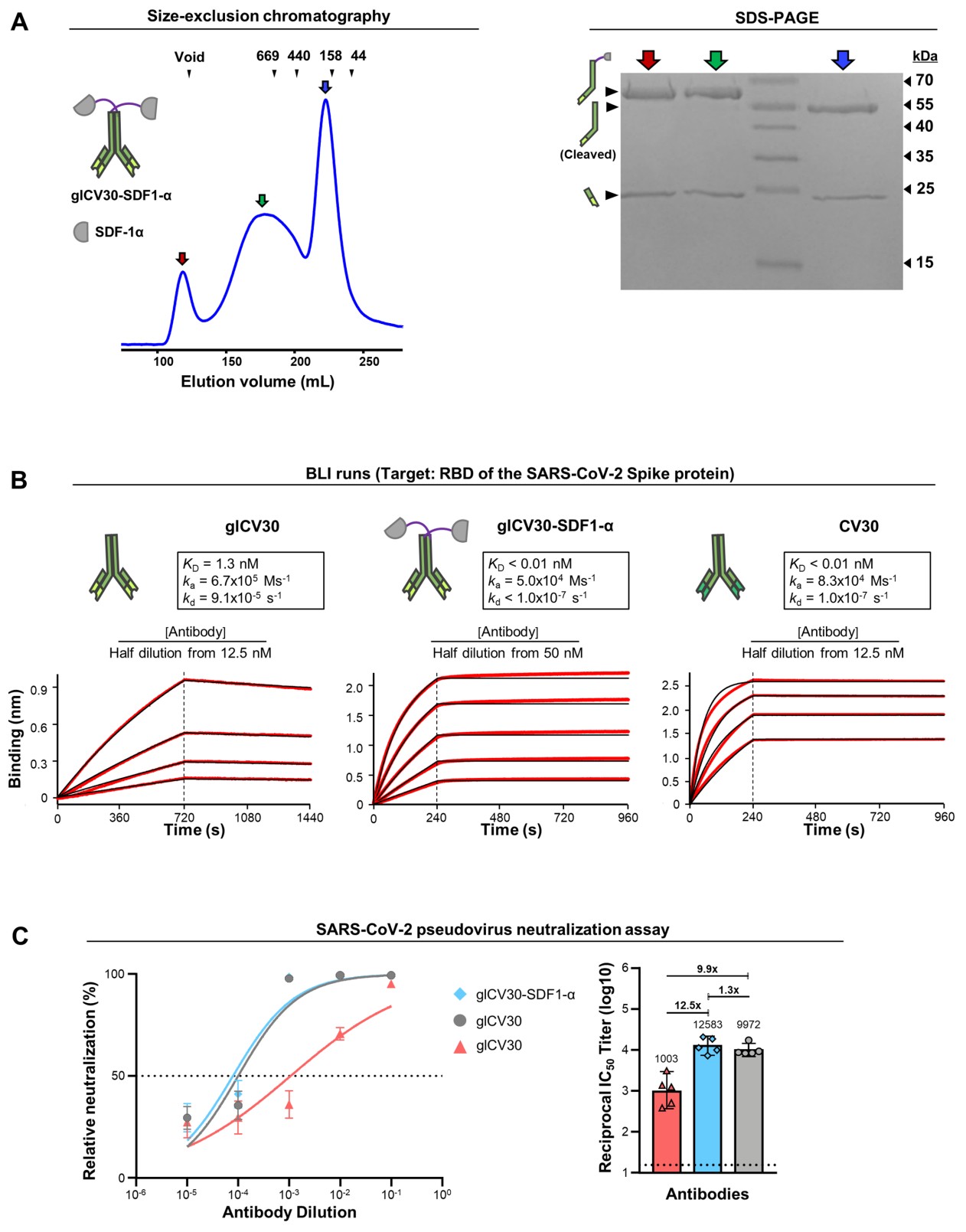

**Figure 7.** Analyses of glCV30-SDF-1α. (**A**) Elution profile and SDS-PAGE analysis of glCV30-SDF-1α. (**B**) The BLI runs shown on the left of *Figure 5C* are reused here for easy comparison with those for glCV30-SDF-1α. The $K_D$ values could not be accurately determined due to instrumental insensitivity ($K_D$ <10 pM). The affinity-matured version, CV30, was analyzed and its BLI sensorgrams are shown on the right. (**C**) Neutralization of VSV virus pseudotyped with the SARS-CoV-2 spike protein. Each of the three antibodies was serially diluted and added to the rVSV-ΔG-Luc bearing the SARS-

*Figure 7 continued on next page*

*Figure 7 continued*

CoV-2 Spike protein of the Wuhan-Hu-1 strain. The mixture was incubated with HEK293T-hACE2 cells and the luciferase activity was measured. The geometric mean titer (GMT) values with a 95% confidence interval are shown on the right, which corresponds to the $IC_{50}$ values of 0.25, 3.22 and 0.21 µg/ml of CV30, glCV30 and glCV30-SDF-1α, respectively.

The online version of this article includes the following source data for figure 7:

**Source data 1.** SDS-PAGE result for glCV30-SDF-1α.

catenators, both of which are weakly homodimerizing proteins ($K_D$ of 100–150 µM). When fused to antibodies, they resulted in great enhancement of the effective antigen-binding avidity of the antibodies, which was due to drastically reduced rate of dissociation of the fused antibody molecules from the immobilized antigens.

The 'antibody catenation on a target surface' method presented herein might find practical applications. First, it can be applied to therapeutic antibodies against viruses, which have multiple copies of target antigens on their surface. Second, it can be used for sandwich-type point-of-care biosensors in which a second antibody is catenated to increase the sensitivity of detection. Third, this method can be used to sense biomarkers that exist in a very low number on a target cell (e.g. copy number <10), which requires an extremely high-binding avidity of a probe antibody. For this application, employing an antibody with high antigen-binding affinity (e.g. $K_D$ <1 nM) and a catenator with high homodimerization affinity (e.g. $(K_D)_{catenator}$ <1 µM), would be necessary to overcome the low proximity effect due to the scarcely present antigen molecules. Fourth, it might also be applied to antibody-based targeted cancer therapy where side effects arising from antibody binding to normal cells are a general problem. Since cancer-associated antigen molecules are lower in number on normal cells than they are on cancer cells, catenated anticancer antibodies would be concentrated on the surface of cancer cells, because the effective binding avidity of a $^{cat}Ab$ depends on the number of antigen molecules on a target surface. In particular, this approach would greatly reduce the intrinsic toxicity of antibody-drug

**Table 1.** Simulation specifications.
The definition and values of the parameters used in the presented simulations are tabulated.

| Parameters | Description | Values |
|---|---|---|
| **Specification of $^{cat}Ab$** | | |
| $K_D$ | Dissociation constant of antibody | 10 nM |
| $(K_D)_{catenator}$ | Dissociation constant of catenator | 10 nM-10 mM |
| $[^{cat}Ab]$ | Antibody concentration | 1 nM |
| $l$ | Length of the flexible linker | 6 nm |
| $c$ | Length of the catenator | 2 nm |
| $L$ | Reach length ($l+c/2$) | 7 nm |
| **Specification of the target surface** | | |
| $N_{total\_binding\_sites}$ (in **Figures 3 and 4**) | Number of antibody-binding sites | 98–102 |
| Connectivity number (in **Figures 3 and 4**) | Number of possible catenation | 3 (Hexagonal) 4 (Square) 6 (Triangular) |
| $d$ (in **Figures 3 and 4**) | Distance between adjacent binding sites | 12 nm |
| $L_{surface}$ (in **Figure 4**) | Surface area of the target surface | 40 nm$^2$ |
| Binding site density (in **Figure 4**) | Surface density of the binding sites | 1.47–11.76 (per 7x7 nm$^2$) |
| **Specification of simulation** | | |
| Updates/MCMC step | Number of updates in one MCMC step | 30,000–100,000 |
| Sampling size | Number of sampling for a parameter set | 1024 |

conjugates that are widely used currently. Of note, a catenator fused to the C-terminus of Fc would not affect the effector function Fc through the Fcγ receptor-binding site and FcRn binding site on it.

While we demonstrated that dual catenator-fused heterodimeric IgGs can enhance binding avidity, a higher (or at least the same) catenation effect should be observed for the conventional homodimeric IgGs. To prevent the oligomer formation of the homodimeric IgGs or potential intramolecular homodimerization of the catenator, a more robust catenator has to be employed. Specifically, the ideal catenator should geometrically disallow intramolecular homodimerization, exhibit fast association kinetics, and be able to withstand the standard low pH purification step. On the other hand, our demonstration indicates that this approach can be applied to bispecific antibodies employing a heterodimeric Fc.

## Materials and methods
### MCMC simulation

Simulation runs were carried out in the three steps stated below with specification of the target surface, $K_D$ (for antibody-antigen interaction), $(K_D)_{catenator}$ (for catenator-catenator interaction) and $f(d)$ (effective local concentration of the catenator). In all simulations, the number of $^{cat}Ab$ was far more than that of the binding sites, and therefore, the concentration of free $^{cat}Ab$ was assumed to be the same as that of total $^{cat}Ab$ (free $^{cat}Ab$ +antigen-bound $^{cat}Ab$). Simulation parameters their set values are listed in *Table 1*.

### Step 1. Initialization step

1. A specified 3D target surface is implemented by assigning binding sites to specific locations on the surface.
2. Each binding site is set to be unoccupied.

### Step 2. *MCMC stochastic update* step

The following sub-steps (1-3) are iterated sufficient times to ensure thermodynamic equilibration.

1. A random binding site *BS1* is chosen from the target surface.
2. The binding status of *BS1* is updated.
   If *BS1* is unoccupied, its status is changed to the occupied status with the acceptance probability of $\max(1, \frac{[^{cat}Ab]}{K_D})$.
   If *BS1* is occupied, its status is changed to the unoccupied status with the acceptance probability of $\max(1, \frac{K_D}{[^{cat}Ab]})$.
3. An occupied binding site *BS2* right next to *BS1* is picked on the target surface, and the catenation status of the pair (*BS1*, *BS2*) is updated.

   If (*BS1*, *BS2*) is uncatenated, and if both *BS1* and *BS2* have an unengaged catenator, its status is changed to the catenation status with the acceptance probability of $\max(1, \frac{f(d)}{(K_D)_{catenator}})$.

   If (*BS$_1$*, *BS$_2$* is catenated, its status is changed to the uncatenated status with the acceptance probability of $\max(1, \frac{(K_D)_{catenator}}{f(d)})$

### Step 3. Sampling step

1. The update step is stopped, and the final status of the target surface is recorded.
2. The total number of occupied and unoccupied binding sites are counted.

The codes for the model system and simulations are available in MATLAB and available on Github (copy archived at *Song, 2020*). A detailed description is provided in Readme.

### Preparation of antibodies and catenator-fused antibodies

For preparing SAM-Trz(2 m)/HetFc-SDF-1α, DNA fragments encoding the two heavy chains containing the knobs mutations or the holes mutations (*Leaver-Fay et al., 2016*) were synthesized (IDT) and cloned into the pCEP4 vector (Invitrogen). The final cloned vectors were to express the knob H chain fused to SAM-(His)$_6$, and the hole H chain fused to SDF-1α-MBP. The two vectors were amplified using the NucleoBond Xtra Midi kit (Macherey-Nagel) and introduced into the CHO-S cells (Gibco) together

with the light chain-encoding vector. The transfected cells were grown in the ExpiCHO expression medium (Gibco) for seven days post-transfection. Cell cultured media were collected by centrifugation at 4 °C, filtered through 0.45 µm filters (Millipore), and loaded onto Ni-NTA resin (Thermo Scientific) and subsequently onto amylose resin (NEB). The MBP tag was cleaved by TEV protease. The MBP was intentionally used to remove homodimeric fraction. The antibody was further purified using a HiLoad 26/60 Superdex 200 gel-filtration column (Cytiva) equilibrated with a buffer solution containing 20 mM Tris-HCl (pH 7.5) and 150 mM NaCl. SAM-Trz(3 m)/HetFc-SDF-1α, SAM-Obinutuzumab(Y101L)/HetFc-SDF-1α and SAM-glCV30/HetFc-SDF-1α were prepared similarly.

For preparing the homodimeric glCV30-SDF-1α, each DNA fragment encoding heavy chain variable regions ($V_H$) and light chain variable regions ($V_L$) of glCV30 were synthesized (IDT) and cloned into the pCEP4 vector. DNA fragments of $C_{H1}$-$C_{H2}$-$C_{H3}$ of the gamma heavy chain and $C_L$ of the kappa-type light chain were inserted into the pCEP4 vector encoding $V_H$ or $V_L$, and the resulting vectors were named glCV30 Hc and glCV30 Lc, respectively. DNA fragment encoding SDF-1α was synthesized (IDT) and cloned into the glCV30 Hc next to $C_{H3}$ of glCV30 with $(G_4S)_2$ linker sequence (glCV30-SDF-1α Hc). The glCV30-SDF-1α Hc and glCV30 Lc vectors were introduced into the CHO-S cells (Gibco). The transfected cells were grown in the ExpiCHO expression medium (Gibco) for ten days post-transfection. The culture supernatant was diluted by the addition of a binding buffer (150 mM NaCl, 20 mM $Na_2HPO_4$, pH 7.0) to a 1:1 ratio, loaded onto an open column containing Protein A resin (Sino Biological), and eluted with an elution buffer (0.1 M glycine, pH 3.5). The eluent was immediately neutralized by a neutralizing buffer (1 M Tris-HCl, pH 8.5), and the antibodies were further purified using a HiLoad 26/60 Superdex 200 gel-filtration column. The cloning, protein production, and purification procedures for mother antibodies were virtually identical to those used for glCV30-SDF-1α.

## Bio-layer interferometry

BLI experiments were performed to measure dissociation constants using an Octet R8 (Sartorius). Biotinylated SARS-CoV-2 RBD (Acrobio system) or biotinylated Her2/ERBB2 (Sino Biological) was loaded to a streptavidin biosensor tip (Sartorius) for 120s or 180s. A baseline was determined by incubating the sensor with Kinetics Buffer (Sartorius) for 60s. Antibody samples at different concentrations went through the association phase for 240s or 480s, and the dissociation phase for 720s or 1080s. All reactions were carried out in the Kinetics Buffer (Satorius). The binding kinetics were analyzed using the Octet DataAnalysis 10.0 software (Sartorius) to deduce the kinetic parameters. Experiments were performed in triplicate for glCV30 and glCV30-SDF-1α and duplicate for Trastuzumab(N30A/H91A) and Trastuzumab(N30A/H91A)-SDF-1α.

## Cell-binding assay

Flow cytometry experiments were performed to compare the binding efficiencies of Obinutuzumab(Y101L) and SAM-Obz(Y101L)/HetFc-SDF-1α. The SU-DHL5 cell line (DSMZ) was used for the experiments. Cells were cultured in RPMI media (Sigma-Aldrich) containing 10% fetal bovine serum at 37 °C and under 5% $CO_2$. The cells were subcultured every 2–3 days to maintain >95% cell viability. After centrifugation, the cell pellet was resuspended in the PBS buffer containing 1% BSA (PBSF), and more than $2x10^5$ cells per well were plated in 96-well round bottom plates containing 200 µL media. The cells were treated with the antibody samples at room temperature for 1 h, and washed three times using PBSF and resuspended in 200 µL of PBSF. Each well was analyzed using Cytoflex_Plate Loader (Beckman Coulter). The sample flow rate was 30 µL/min, and 15,000 cells were counted per well.

## Pseudovirus neutralization assay

To prepare of VSV pseudotyped with the SARS-CoV-2 Spike protein of the Wuhan-Hu-1 strain, HEK293T cells (ATCC) plated overnight previously at $3x10^6$ cells in a 10 cm dish were transfected using calcium phosphate with 15 µg plasmid encoding the spike protein of SARS-CoV-2 with 18-residue deletion at the cytoplasmic tail. At 24 hr post-transfection, cells expressing the Spike protein were infected for 1 hr with recombinant VSV, in which G gene was replaced with a luciferase gene (rVSV-ΔG-Luc). The cells were washed three times with Dulbecco's phosphate-buffered saline, and 7–10 mL of the same media with 10% FBS was added. At 24–48 hr post-infection, the culture media were harvested, filtered with a 0.45 µm filter, and then stored at –80 °C for neutralization assay. For the pseudovirus neutralization assay, serially diluted CV30, glCV30 and

glCV30-SDF-1α antibodies were mixed with the pseudovirus solution for 1 hr, and the mixture was added to HEK293T cells expressing human ACE2 (HEK293T-hACE2; 3x104 cells per well in 96-well plate), which were previously seeded overnight. Cells were lysed with Passive Lysis Buffer (Promega, E1941) At 24 hr post-transfection, LAR II (Promega, E1501) was added to the lysate, and the luciferase activity was measured. The percent neutralization was normalized to uninfected cells (100% neutralization) and infected cells (0% neutralization), both in the absence of antibody. The IC50 titers were determined with the nonlinear curves of the log(antibody) versus normalized response using Prism v9 (GraphPad).

### Cell lines
The CHO-S (Gibco), SU-DHL5 (DMSZ), and HEK-293T (ATCC) cell lines were negative for mycoplasma. Additionally, the identity of the SU-DHL5 and HEK-293T cell lines was confirmed by Short Tandem Repeat profiling.

### Figure preparation
The computational models of an antibody and an antibody-catenator in *Figure 1A* were generated by using the ROSETTA software (*Leman et al., 2020*), and are presented by PyMOL (*Delano, 2004*).

## Acknowledgements

This work was supported by the Samsung Research Funding & Incubation Center of Samsung Electronics under Project Number SRFC-MA2002-06.

---

## Additional information

### Competing interests
Jinyeop Song, Bo-Seong Jeong, Seong-Woo Kim, Seong-Bin Im, Byung-Ha Oh: co-inventors in a patent application covering the antibody catenation method presented in this article (10-2023-0067544, Republic of Korea). The other authors declare that no competing interests exist.

### Funding

| Funder | Grant reference number | Author |
| --- | --- | --- |
| Samsung | SRFC-MA2002-06 | Byung-Ha Oh |

The funders had no role in study design, data collection and interpretation, or the decision to submit the work for publication.

### Author contributions
Jinyeop Song, Conceptualization, Data curation, Software, Formal analysis, Validation, Visualization, Methodology, Writing – original draft, Writing – review and editing; Bo-Seong Jeong, Conceptualization, Data curation, Validation, Investigation, Visualization, Methodology, Writing – original draft, Writing – review and editing; Seong-Woo Kim, Investigation, Visualization, Writing – review and editing; Seong-Bin Im, Investigation, Writing – review and editing; Seonghoon Kim, Jae U Jung, Data curation; Chih-Jen Lai, Data curation, Formal analysis; Wonki Cho, Conceptualization, Writing – original draft, Writing – review and editing; Myung-Ju Ahn, Conceptualization, Writing – review and editing; Byung-Ha Oh, Conceptualization, Resources, Data curation, Supervision, Funding acquisition, Visualization, Writing – original draft, Project administration, Writing – review and editing

### Author ORCIDs
Jinyeop Song ⓘ http://orcid.org/0000-0002-4113-5185
Bo-Seong Jeong ⓘ http://orcid.org/0000-0003-0629-6722
Seong-Woo Kim ⓘ http://orcid.org/0000-0002-4374-2093
Seong-Bin Im ⓘ http://orcid.org/0000-0003-1488-6386
Byung-Ha Oh ⓘ http://orcid.org/0000-0002-6437-8470

**Decision letter and Author response**
Decision letter https://doi.org/10.7554/eLife.81646.sa1
Author response https://doi.org/10.7554/eLife.81646.sa2

---

## Additional files

### Supplementary files
- MDAR checklist
- Source code 1. Github repository to generate simulation data for *Figures 3 and 4*.

### Data availability
MATLAB scripts reproducing all simulation result in the paper are provided as Source code 1 (and are also publicly available through a MIT License on GitHub, copy archived at *Song, 2022*). All simulation data generated by the scripts are provided as Source data.

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
