## [Editor Report]

This important study documents the use of computational models and protein design to enhance antibody binding. The new method could have a broad and immediate impact on a variety of diagnostic procedures that use antibodies as sensitivity is often an issue in these kinds of experiments. The evidence produced is highly compelling through demonstration of the substantial sensitivity enhancement achieved in two test cases. This manuscript will likely be of interest to researchers who use antibodies for diagnostic and therapeutic purposes.

---

## [Decision Letter]

**Decision letter after peer review:**

Thank you for submitting your article "Noncovalent antibody catenation on a target surface drastically increases the antigen-binding avidity" for consideration by *eLife*. Your article has been reviewed by 3 peer reviewers, and the evaluation has been overseen by a Reviewing Editor and Tony Ng as the Senior Editor. The reviewers have opted to remain anonymous.

Essential revisions:

Regarding experiments:

1) There needs to be a negative control for the affinity measurements, such as incorporating a dimerization-defective variant of the SDF-1alpha catenator and seeing if the avidity effect disappears.

2) The experimental assessment focuses solely on the binding affinity and no other aspects of the antibodies, such as purity and solution properties. If the catenator negatively impacts purification, that is information that should be included in the manuscript. The behavior of the antibodies in the solution is particularly important. You want the catenation to be highly dependent on the presence of the target antigen. The catenator could induce the antibodies to form an essentially infinite array and/or large aggregates. Aggregates tend to be highly immunogenic, so if any potential targets of this method are intended to be injected into animals or people, unwanted immune responses could be a serious issue. Gel filtration and dynamic light scattering are two potential methods that can assess the size of the antibodies +/- catenator and +/- antigen to ensure that the antibodies exhibit the desired behavior.

3) It should be stated that the two antibodies, Trastuzumab(N30A/H91A) and glCV30, were selected for weakened antigen-binding affinities.

4) On page 12, line 246, SDF-1α KD was 150 uM; on page 14, line 299, SDF-1α KD was 150 nM. Which one is correct?

5) What is the physiological expression and function of SDF-1α? Would injection of IgG fused with SDF-1α disturb the physiological function of SDF-1α?

6) What are the expression yields of catenator-fused antibodies? How do the yields compare to those of the unmodified antibodies?

7) It would greatly improve the implications of IgG catenation if relevant antibody functional assays can be performed for the two tested antibodies. For example, an in vitro assay to compare the function of Trastuzumab(N30A/H91A), cat-Trastuzumab(N30A/H91A), and Trastuzumab. Likewise, a SARS-CoV-2 neutralization assay to compare glCV30, cat-glCV30, and CV30.

Regarding computations:

8) The simulations could be improved. The third major step in the simulations seems to favor the binding ability of catenated IgGs. If so, in any means the simulations will yield a higher binding affinity for the catenated homodimeric IgG. This should be clarified.

9) What is the role of the concentration of IgGs on the binding behavior? Would it be possible somehow to include this in the simulations and also link it to the experiments?

10) While the results render the enhanced binding affinity of the catenated homodimeric IgGs, the study would benefit from a more elaborated interpretation and discussions of the results.

(10a) One interesting base of discussion may include how the fusion of the catenator may likely affect the binding behavior, the intrinsic binding behavior, and/or on the global structural changes of IgGs per se, beyond its proximity-driven contribution. Please refer both to the monomeric and homodimeric (catenated) forms. Would it lead to a more restricted structure in the mobility in the unbound states so as to decrease the entropic cost for the binding and thus increase the binding avidity/affinity (in addition to external proximity-driven association)? In other words, what would be the role of entropy in the free energy of binding, given that the enthalpic contributions remain the same? Possible effects of the length of the catenator should also in parts be related to the entropy. For example, if a longer and more flexible catenator is considered, what would the resulting observation be? Both experimentally and computationally.

(10b) On the other side, simple simulation approaches have a high value with a level of abstraction while still keeping the physical and biological relevance. In the simulations, i.e., in the sampling of various states, three main terms/rules to govern the behavior are implemented. One is a term favoring an increase in the ability to bind (preventing unbinding) to the surface upon the catenation of IgGs. This may need to be substantiated for the simulations not imposing a presumed ability to increase the binding (or decrease the unbinding) ability upon catenation.

(10c) The weakly homodimerizing state of the catenator appears as one of the important aspects of the proposed design strategy. Would it also be possible that the experimental observations may readily also imply the higher binding ability of the catenator fused IfgG without the homodimerization on the surface (due to the reduced entropic cost for the binding)? The presentation of the evidence of the homodimerization of the catenator and the catenated IgGs on the surface would strengthen the findings and discussions.

---

## [Author Response]

1) There needs to be a negative control for the affinity measurements, such as incorporating a dimerization-defective variant of the SDF-1alpha catenator and seeing if the avidity effect disappears.

Employing the HetFc was quite instrumental in making a control construct, which is an antibody-catenator with only one arm, which cannot be catenated;

**Author response image 1. sa2fig1:** 

Consistently, the avidity enhancement was not observed with this construct. The results are shown as Figure 6A in the revision.(*Both the single and the dual catenator-fused antibodies have mScarlet on the C-terminus of the light chain. The mScarlet fusion is for later cell-based experiments.)

2) The experimental assessment focuses solely on the binding affinity and no other aspects of the antibodies, such as purity and solution properties. If the catenator negatively impacts purification, that is information that should be included in the manuscript. The behavior of the antibodies in the solution is particularly important. You want the catenation to be highly dependent on the presence of the target antigen. The catenator could induce the antibodies to form an essentially infinite array and/or large aggregates. Aggregates tend to be highly immunogenic, so if any potential targets of this method are intended to be injected into animals or people, unwanted immune responses could be a serious issue. Gel filtration and dynamic light scattering are two potential methods that can assess the size of the antibodies +/- catenator and +/- antigen to ensure that the antibodies exhibit the desired behavior.

The revised manuscript includes Figure 5B and C, Figure 6A and Figure 6—figure supplement 1, which show the purification process for the newly generated antibody-catenator constructs. The elution profiles from a size-exclusion column confirm that these constructs were monomeric in solution.

3) It should be stated that the two antibodies, Trastuzumab(N30A/H91A) and glCV30, were selected for weakened antigen-binding affinities.

We state that “The two antibodies were chosen based on their weak antigen-binding affinities.” (page 13, in the revision).

4) On page 12, line 246, SDF-1α KD was 150 uM; on page 14, line 299, SDF-1α KD was 150 nM. Which one is correct?

Thank you for the opportunity to correct our typographical error. 150 μM is correct, and nM is now corrected to μM.

5) What is the physiological expression and function of SDF-1α? Would injection of IgG fused with SDF-1α disturb the physiological function of SDF-1α?

SDF-1α belongs to the chemokine family, members of which activate leukocytes and are induced by proinflammatory stimuli. The first 7 N-terminal residues are known to serve as a receptor-binding site for CXCR4 [EMBO J. 1997;16:6996–7007]. It should be confirmed that SDF-1α or its variants (e.g., the N-terminal deletion mutant) do not cause a side effect, if SDF-1α would be used for therapeutic purposes.

6) What are the expression yields of catenator-fused antibodies? How do the yields compare to those of the unmodified antibodies?

On our hands, the total yield of the dual catenator-fused antibodies (~5 mg from 25 mL culture) was about half of the production of the mother antibodies. Usually, fusing a protein to IgG antibody reduces the production of the fused antibody, but the yield quite depends on the protein being fused, as far as I know.

7) It would greatly improve the implications of IgG catenation if relevant antibody functional assays can be performed for the two tested antibodies. For example, an in vitro assay to compare the function of Trastuzumab(N30A/H91A), cat-Trastuzumab(N30A/H91A), and Trastuzumab. Likewise, a SARS-CoV-2 neutralization assay to compare glCV30, cat-glCV30, and CV30.

We conducted a cell-binding experiment using flow cytometry, which demonstrates that SAM-Obinutuzumab(Y101L)/HetFc-SDF-1α binds to CD20-expressing cells (SU-DHL5 cell line) much more effectively than the mother antibody (Figure 6B, in the revision). A new author, Seonghoon Kim, was involved in the extension of our work to include Obinutuzumab.

In addition, pseudovirus (instead of SARS-CoV-2) neutralization assay was performed exactly as the reviewer commented, but with the old construct (glCV30-SDF-1α). glCV30-SDF-1α exhibited a ~15 fold lower IC_50_ value in comparison with glCV30, which is comparable to that of CV30, having an extremely potent antigen-binding affinity (*K*_D_ of < 0.01 nM). The result is now shown as Figure 7B. The description of this assay and the results are now stated under the subtitle of “Incidental observations” (pages 15-16, in the revision). The experimenters (Jae U. Jung and Chih-Jen Lai at the Cleveland Clinic) are included as authors in the revised version.

Regarding computations: (8) The simulations could be improved. The third major step in the simulations seems to favor the binding ability of catenated IgGs. If so, in any means the simulations will yield a higher binding affinity for the catenated homodimeric IgG. This should be clarified.

We agree with the review in that the third rule favors the binding ability of catenated IgGs, because it assumes that catenated antibodies are not allowed to dissociate from the binding site. While this assumption is not exactly correct, we think that it is valid, considering the behavior of a multivalent ligand. When the IgG portion dissociates completely from the binding site, it is still anchored by the catenation arm, and thus it will rebind the same binding site immediately. This postulation is supported by binding ΔG calculation presented as a response to comment 10b below, and agrees with the quantitative analysis showing that multivalent ligand exhibits orders of magnitude binding likelihood increase when the ligand size is comparable to the stretch length of a conjugating linker [Liese, S. and Netz, R. R., *ACS Nano,* 12**,** 4140 (2018)].

9) What is the role of the concentration of IgGs on the binding behavior? Would it be possible somehow to include this in the simulations and also link it to the experiments?

The binding site occupancy depends on [^cat^Ab]/*K*_D_. Figure 4—figure supplement 2 shows the binding site occupancy and (*K*_D_)_eff_ as a function of (*K*_D_)_catenator_. In this simulation, [^cat^Ab] was fixed (10^-9^ M) while *K*_D_ was varied (from 10^-8^ to 10^-6^). In the figure legend and in the main text, we now explicitly state that *K*_D_ was varied from 10^-8^ to 10^-6^ (pages 12, 35 in the revision). To address this comment, we set *K*_D_ = 10 nM (as used for simulation in Figures 3 and 4), and varied [^cat^Ab] from 0.1 to 10 nM. The binding site occupancy and (*K*_D_)_eff_ as a function of [^cat^Ab] are plotted for three different set values of (*K*_D_)_catenator_ (1 μM, 10 μM and 100 μM). The new figures are now presented as Figure 4—figure supplement 3. This simulation shows that the enhancement of (*K*_D_)_eff_ by increasing the concentration of ^cat^Ab is much less dramatic than that by increasing the affinity for catenator homodimerization. A new figure legend and describing sentences in the main text are added (page 12, in the revision).

10) While the results render the enhanced binding affinity of the catenated homodimeric IgGs, the study would benefit from a more elaborated interpretation and discussions of the results.

We have demonstrated the binding avidity enhancement by employing a heterodimeric Fc. Accordingly, the following discussion is now stated in the revision (pages 19-20, in the revision);

“While we demonstrated that dual catenator-fused heterodimeric IgGs can enhance binding avidity, the oligomer formation or potential intramolecular homodimerization of the catenator necessitates the development of a more robust catenator for application to conventional homodimeric IgGs. Specifically, the ideal catenator should geometrically disallow intramolecular homodimerization, exhibit fast association kinetics, and be able to withstand the standard low pH purification step. On the other hand, our demonstration indicates that this approach can be applied to bispecific antibodies employing a heterodimeric Fc.”

(10a) One interesting base of discussion may include how the fusion of the catenator may likely affect the binding behavior, the intrinsic binding behavior, and/or on the global structural changes of IgGs per se, beyond its proximity-driven contribution. Please refer both to the monomeric and homodimeric (catenated) forms. Would it lead to a more restricted structure in the mobility in the unbound states so as to decrease the entropic cost for the binding and thus increase the binding avidity/affinity (in addition to external proximity-driven association)? In other words, what would be the role of entropy in the free energy of binding, given that the enthalpic contributions remain the same? Possible effects of the length of the catenator should also in parts be related to the entropy. For example, if a longer and more flexible catenator is considered, what would the resulting observation be? Both experimentally and computationally.

To fully address this comment, we would need to consider the detailed molecular behavior of the IgG part, catenator and linker, probably using molecular dynamics simulation, which we think is outside the scope of the current work. We like to qualitatively describe what we think about the raised issues. Fused to the C-terminus of Fc, the catenator won’t affect the complementary determining region (CDR) of Fab which is located on the opposite side of the C-terminus of Fc. This notion is supported by the observation that all catenator-fused antibodies exhibited association kinetics similar to those of the mother antibodies (Figure 5).

Regarding the mobility of the structure, we presume that the fused catenator would not interact with the antibody portion and thus it would not affect the intrinsic structural mobility of the antibody.

Since the catenator is fused to the C-terminus of Fc by a flexible linker, the homodimerization of catenator would decrease the entropy upon catenation. However, the enthalpic contribution would overcome the entropic loss, and result in negative free energy of the catenator homodimerization.

Figure 2—figure supplement 1 (in the revision) shows the simulation for five different values of the reach length (*R*), which is the sum of the linker length and a half of the catenator length. The simulation results show that the likelihood of catenation decreases as the linker length increases over the distance (*d*) between the two adjacent ^cat^Ab-2Ag complexes, while it is maximum when the reach length equals *d*. Since the catenator length is fixed, increasing the linker length (such that *R* > *d*) will lower the catenation effect.

(10b) On the other side, simple simulation approaches have a high value with a level of abstraction while still keeping the physical and biological relevance. In the simulations, i.e., in the sampling of various states, three main terms/rules to govern the behavior are implemented. One is a term favoring an increase in the ability to bind (preventing unbinding) to the surface upon the catenation of IgGs. This may need to be substantiated for the simulations not imposing a presumed ability to increase the binding (or decrease the unbinding) ability upon catenation.

This comment is nearly the same as comment 8 above about the restricted dissociation of catenated antibody molecules from the binding site. We did rough calculations of entropy changes in the binding of uncatenated antibody and catenated antibody to the binding site. The largest difference in entropic change originates from a change in configurational entropy, *i.e*, by the change in volume that an anibody molecule can sample.

(1) Entropy of an uncatenated antibody:

A free antibody can sample an entire volumetric space. The per-particle entropy of N identical molecules in volume V is calculated as: su=SN=kB[ln⁡(VN)+32ln⁡(2πh02mkBT)+52] Therefore, the per-particle entropy is a function of [^cat^Ab] and temperature T.su([catAb], T)=kB[ln⁡(1[catAb]NA)+32ln⁡(2πmkBTh02)+52] (2) Entropy of a catenated antibody :

A catenated Ab has smaller configurational entropy as it is anchored by the linker and thus confined in a smaller space. We approximate the sample space as a sphere with the radius twice the stretched reach length (2L). Additional entropic effect by the linker is neglected, since the IgG portion has a much larger molecular mass. Then the entropy of a catenated antibody is as follows.

sc=kBln⁡[V∗(2πmkT)32 h03]=kB[ln⁡(32π3L3)+32ln(2πmkBTh02)] Under the simulation scheme, for KD=10 nM, [catAb]=1 nM, T=300 K and l=5−15 nm, *T*Δ*S* as a function of linker length was calculated and a plot is shown in Author response image 2.

Under the assumption that enthalpic contributions remain the same, the free energy changes for antibody (Ab) are (Gbound\ Ab−Gfree\ Ab)=−kBNATln(KD)=1.37kcalmol (uncatenated)

(Gbound\ Ab−Gfree\ Ab)=−kBNATln(KD)+kBNAT(sc−su)=−7.30kcalmol (Catenated)

Therefore, about 10% of monomeric antibody molecules bind to the binding site, whereas more than 99% of catenated antibody molecules bind to the binding site. The above calculation implies that configurational entropic change is sufficient to account for the order of magnitude difference in the binding avidity between monomeric antibody and catenated antibody in our simulations.

(10c) The weakly homodimerizing state of the catenator appears as one of the important aspects of the proposed design strategy. Would it also be possible that the experimental observations may readily also imply the higher binding ability of the catenator fused IfgG without the homodimerization on the surface (due to the reduced entropic cost for the binding)? The presentation of the evidence of the homodimerization of the catenator and the catenated IgGs on the surface would strengthen the findings and discussions.

This comment is in line with comment (2). We constructed SAM-Trastuzumab(N30A/H91A/Y100A)/HetFc having one catenator arm. This antibody did not show avidity enhancement, in contrast with SAM-Trastuzumab(N30A/H91A/Y100A)/HetFc-SDF-α, having two catenator arms. The results are shown in Figure 6A in the revision.